

# Sex-based influence of morphological and physical attributes on serve performance in adolescent volleyball players

Rani Asfour[1], Hamza Marzouki[2], Ibrahim Ouergui[2], Jabeur Methnani[3], Khalifa Al-Jadidi[4], Majid Al-Busafi[4], Yung-Sheng Chen[5], Nabil Gmada[4] and Ezdine Bouhlel[3]

[1] High Institute of Sport and Physical Education of Ksar-Said, Tunis, Tunisia
[2] High Institute of Sport and Physical Education of Kef, University of Jendouba, Tunisia; Research Unit: Sport Sciences, Health and Movement, University of Jendouba, Kef, Tunisia
[3] Research Laboratory, Exercise Physiology and Physiopathology: from Integrated to Molecular "Biology, Medicine and Health" (LR19ES09), Faculty of Medicine of Sousse, University of Sousse, Sousse, Tunisia
[4] Physical Education and Sport Sciences Department, College of Education, Sultan Qaboos University, Muscat, Oman
[5] Department of Exercise and Health Sciences, University of Taipei, Exercise and Health Promotion Association, New Taipei City, and High Performance Unit, Chinese Taipei Football Association, Taipei, Taiwan

Corresponding author
Nabil Gmada, ngmada@squ.edu.om

## ABSTRACT

This study examined sex differences in anthropometric and physical attributes, and their relationships with volleyball serve accuracy and speed among adolescent volleyball players. Fifty athletes (age: $13.1 \pm 0.4$ years; peak height velocity: $-2.2 \pm 0.3$ years) voluntarily participated in this study. Anthropometric attributes, physical performances, and technical performance (*i.e.*, serve accuracy and speed) were measured. The result showed that boys were taller, leaner, and had greater leg length than girls (all $p < 0.001$). Boys also outperformed girls in change of direction ($p < 0.01$), vertical jump ($p < 0.0001$), upper limb strength ($p < 0.005$), balance, and serve speed (both $p < 0.0001$). Height, arm length, lower limb length, and hand length were positively correlated with serve accuracy and speed for both boys and girls (range: $r = 0.497$ to $0.789$; all $p < 0.05$). Serve accuracy and speed were associated with all physical performance measures (range: $r = 0.402$ to $0.908$; all $p < 0.05$), except 10-meter sprint time, which was only significantly correlated with serve speed in girls ($r = -0.403$; $p < 0.05$). The main anthropometric factor influencing serve accuracy was height in boys ($R^2 = 0.901$; $p < 0.0001$) and hand length in girls ($R^2 = 0.650$; $p < 0.0001$); the opposite was true for serve speed. For physical performance, change of direction, flexibility, and balance were the main factors influencing accuracy in boys ($R^2 = 0.717$; $p < 0.0001$) and girls ($R^2 = 0.820$; $p < 0.0001$). Serve speed was influenced by hand grip force, upper limb strength, and balance in boys ($R^2 = 0.770$; $p < 0.0001$), while hand grip force was the main factor in girls ($R^2 = 0.722$; $p < 0.0001$). In conclusion, body span and physical abilities, such as balance, flexibility, and strength are essential for volleyball serve accuracy and speed in adolescent players.

## INTRODUCTION

Volleyball is a popular team sport that requires a distinct combination of high physical aptitude and several technical and tactical aspects for sports achievement (*Sheppard et al., 2009*). The game requires players to perform explosive actions accurately, such as serving, jumping, spiking, and sprinting (*Tsoukos et al., 2019*). Understanding the various factors contributing to volleyball players' performance can provide valuable insights for coaches, athletes, and sports scientists. Morphological characteristics, physical capacities, and technical skills are among the key elements influencing success in volleyball, which can vary significantly between individuals and across sexes.

Moreover, performance in volleyball actions such as the serve and spike depend on the interaction between biomechanical, neuromuscular, and morphological factors. Effective force transmission to the ball requires a well-coordinated kinetic chain, involving the sequential activation and alignment of the hips, trunk, shoulders, elbows, and wrists (*Baena-Raya et al., 2021*; *Wagner et al., 2011*). Key biomechanical variables such as shoulder–hip separation, angular velocities of the hip and shoulder, and joint angles at ball impact are strongly associated with ball speed (*Oliveira et al., 2020*). Longer limbs can enhance end-point velocity, thereby improving serve performance (*Van den Tillaar & Ettema, 2004*), while anthropometric features like height and arm span facilitate a higher point of contact and more effective serve angles (*Palao, Manzanares & Valadés, 2014*). In a study comparing elite and sub-elite athletes, *Carvalho, Roriz & Duarte (2020)* reported greater lean mass, body height, and jump capacity in high-level performers. In addition, *Moscatelli et al. (2021)* emphasized the importance of neuromuscular determinants of performance, particularly highlighting transcranial magnetic stimulation as a tool to investigate motor cortex excitability in sports. Together, these findings support the critical role of technical skills, neuromuscular development, and biomechanical factors in enhancing decisive actions such as serving and spiking in volleyball.

Adolescence is a critical period for athletic development, characterized by growth and maturation, hormonal changes, and changes in motor skills (*Da Costa et al., 2023*). This growth period is a critical period for talent identification and athletic development (*Jürimäe, 2018*). In volleyball, chronological differences in body composition, muscle strength, and coordination during this growth period may lead to varied proficiency levels of technical skills (*i.e.,* serving, spiking, and blocking). To our knowledge, only one volleyball study (*Albaladejo-Saura et al., 2022*) reported significant differences in anthropometric and physical fitness attributes between adolescent girls and boys. However, this study did not assess the relationship between anthropometrics, physical fitness, and volleyball performance. Recently, *Pawlik et al. (2022)* assessed the relationships between strength abilities and serve reception efficiency in younger girls (12–13 years) without focusing on the link between morphological and/or physical attributes, and volleyball technique, such as serve efficiency. The authors found significant positive relationships between serve reception efficiency, peak torque ($r = 0.62$) and power of the shoulder joint ($r = 0.58$), medicine ball throw distance ($r = 0.53$), and hand grip force ($r = 0.49$).

The serve, a cornerstone of volleyball, not only initiates play but often determines the game's outcome. Its efficiency is likely influenced by a combination of morphological characteristics such as height and arm span, physical capabilities like strength and power, and technical proficiency. The volleyball serve is a powerful, coordinated ballistic action, involving explosive lower-body strength for jumping, trunk flexibility, balance, upper-limb strength and coordination for arm swing and ball impact. Surprisingly, few studies have explored the relationships between morphological and/or physical fitness factors and technical skills, such as serve efficiency, while considering the athletes' sex. Understanding these interactions and their contribution to serve efficiency in adolescent volleyball players of both sexes could revolutionize training and performance optimization. Sex-specific differences significantly influence volleyball performance and training. Female volleyball players typically display lower upper-body power, which may affect serve and spike velocity (*Reeser et al., 2010*). Sex differences in physical performance begin at the onset of puberty, with a larger rise in circulating testosterone in boys compared to girls. Hormonal changes can impact body composition and physical performances, contributing to higher body size, throwing performance, and isometric strength (*Van den Tillaar & Ettema, 2004*).

While the physiological characteristics of adult volleyball players have been extensively studied, research focusing on adolescent athletes remains comparatively scarce (*Albaladejo-Saura et al., 2022*). Replicating such investigations in younger populations is essential to understand the development of physical performance, morphological attributes, and technical skills during key stages of athletic maturation. These insights can assist coaches in identifying performance limitations and in designing targeted training interventions to optimize physical performance in youth volleyball players.

Thus, our study had two primary objectives. First, we aimed to compare adolescent boys and girls at the same biological age (*i.e.*, pre-peak height velocity (Pre-PHV)) regarding their morphological attributes, physical fitness aptitude, and serve performance in volleyball, with a specific focus on the efficiency of the serve in terms of accuracy and speed. Second, we sought to identify critical determinants of serve performance and potential sex-specific differences that can inform the optimization of training strategies and/or talent selection for young volleyball athletes. First, we hypothesized that there would be significant sex-based differences in morphological characteristics, physical fitness, and serve performance between adolescent boys and girls. We anticipated that boys might demonstrate higher serve speed due to greater strength and power, while girls might exhibit better serve accuracy due to enhanced coordination and technical proficiency at this stage of development. Second, we hypothesized that the relationships between morphological and physical fitness factors and serve efficiency would vary between the sexes, with different key determinants influencing performance for boys and girls.

## MATERIALS AND METHODS

This study examined the relationship between morphological characteristics and physical performance with serve accuracy and velocity in adolescent male and female volleyball players. The protocol testing included 10-m sprint, change of direction (COD), agility

**Table 1 Anthropometric measures in adolescent U14 boys ($n = 25$) and girls ($n = 25$).**

| Variables | Boys | Girls | $p$ | Effect size |
|---|---|---|---|---|
| Age (years)[*] | $13.0 \pm 0.5$ | $13.3 \pm 0.4$ | NS | – |
| PHV (years)[*] | $-2.1 \pm 0.3$[†] | $-2.3 \pm 0.4$ | 0.013 | 0.272 |
| APHV (years)[*] | $15.1 \pm 0.3$ | $15.6 \pm 0.5$[‡] | <0.0001 | 0.519 |
| Height (cm)[*] | $163.6 \pm 3.6$[†] | $157.4 \pm 5.2$ | <0.0001 | 0.570 |
| Body mass (kg)[*] | $50.6 \pm 5.6$ | $49.0 \pm 3.5$ | NS | – |
| BMI (kg m$^{-2}$)[*] | $18.9 \pm 1.9$ | $19.8 \pm 1.5$ | NS | – |
| Body Fat (%)[**] | $25.0 \pm 0.7$ | $26.5 \pm 0.8$[‡] | <0.0001 | 0.805 |
| Arm Length (cm)[**] | $70.0 \pm 6.5$ | $72.0 \pm 9.5$ | NS | – |
| Sitting Length (cm)[*] | $76.7 \pm 1.8$ | $73.6 \pm 3.0$ | NS | – |
| Leg length (cm)[**] | $87.0 \pm 3.5$[†] | $83.0 \pm 3.0$ | <0.0001 | 0.558 |
| Hand length (cm)[**] | $16.0 \pm 2.5$ | $17.0 \pm 2.5$ | NS | |

**Notes.**
[*]Values are expressed as mean $\pm$ SD.
[**]Values are expressed as median $\pm$ interquartile range.
PHV, peak height velocity; APHV, age at peak height velocity; BMI, body mass index.
[†]Significantly different from girls.
[‡]Significantly different from boys.

$T$-test, vertical jump, five jump, seated medicine ball throw, hand grip, sit and reach, Y balance tests, and serve accuracy and velocity tests. Anthropometric measurements included height, body mass, skinfold thicknesses, arm length, hand length, and leg length.

## Participants

A total of fifty adolescent volleyball players (25 girls and 25 boys) selected for the Palestinian National Volleyball Team voluntarily participated in this study (Table 1 for participants characteristics). Maturity status was estimated using the anthropometric-based equation developed by *Mirwald et al. (2002)*, which provides an estimate of the number of years from peak height velocity (PHV). The calculation incorporated variables such as leg length, sitting height, chronological age, body height, and body mass to determine maturity offset. Players were included if they were categorized as pre-PHV (between $-3$ and $-1$ years from PHV), actively engaged in volleyball training and competitions, and free from severe musculoskeletal injuries within the past year, or minor injuries in the month before the assessment. Players were instructed to maintain their regular dietary habits and to refrain from intense exercise 24 h prior to testing. Players and their parents were informed about the testing protocol and the study objectives. Parents signed a written informed consent before the commencement of the study. This study received institutional ethics approval from the Faculty of Medicine of Sousse (Ref: CEFMS 402/2024; date of approval: February 04, 2024), and all procedures were performed in accordance with the Code of Ethics of the World Medical Association and the Declaration of Helsinki.

## Testing protocol

Players performed the tests during three visits during the weekly training program. During the first visit, which occurred 1 week before the beginning of the experimental procedure, players were familiarized with the tests, and anthropometric attributes were

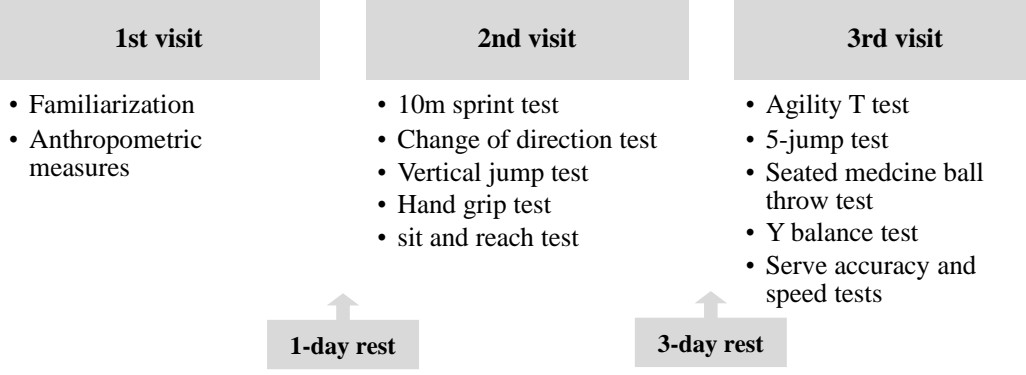

**Figure 1** Schematic representation of the protocol.

assessed. During the second visit, players performed the 10-m sprint, followed by change of direction run, vertical jump, hand grip, and sit and reach tests. During the third visit, athletes performed the agility $T$-test, five jump, medicine ball throw, Y balance tests, and serve accuracy and velocity tests. All tests were administered on three non-consecutive days, allowing recovery to avoid potential fatigue, using the same procedures by the same evaluators, who were not blinded to the players (Fig. 1). During tests, players were provided with the same verbal encouragement to ensure that they performed at maximal effort. The tests were performed on the same indoor court, under similar ambient conditions (temperature: 19–22 °C; relative humidity: 67–76%, measured using Extech device, Nashua, USA), at the same time of day (5:00 pm to 7:00 pm) to avoid any diurnal variation of the performance, and measurements were conducted in the middle of the season (February 2024). In order to prevent the effects of fatigue on subsequent performance, intensive training was avoided 24 h prior to each testing session. Before each experimental session, a standardized warm-up session was performed consisting of 5 min of low intensity running, three lateral runs and three lateral jumps along the net, interspersed by 2 min of passive recovery.

## Parameters
### Anthropometric measurements
Players' body mass was measured using a calibrated digital scale (OHAUS, Florhman Park, NJ, USA) with 0.1 kg precision. Stature was assessed to the nearest 0.1 cm using a portable stadiometer (Seca model 213, Germany). Body mass index (BMI) was calculated by dividing body mass (kg) by the square of body height (m$^2$). Skinfold thicknesses were measured at four sites (*i.e.,* biceps, triceps, subscapular and suprailiac) using a Harpenden caliper (Harpenden/Holtain Calipers, Crosswell, Crymych, Pembrokeshire, UK). Body density was then estimated using the *Durnin & Womersley (1974)* equation, and subsequently converted to body fat percentage using Siri's equation (*1993*). In addition, leg length was measured in centimetres from the anterior superior iliac spine to the most distal part of

the medial malleolus. Hand length was also measured from the tip of middle finger to the tip of radial styloid. Measurements were performed by the same investigator.

### Speed and change of direction assessment

*10-m linear sprint test.* The 10-meter linear sprint test, was assessed using a series of paired photocells (Globus, Mictogate, Bolsano, Italy). Players performed three attempts; the best one was recorded in seconds (s), and retained for analysis. The attempts were separated by 4–5 min recovery intervals (*cf.* Supplementary File for details).

*Change of direction test.* The COD performance was assessed using the $4 \times 10$ m shuttle run test (*Ruiz et al., 2011*). COD was performed with the players starting in a standing position, with their preferred foot forward and placed behind a line traced in the gymnasium. One set of the aforementioned dual-beam electronic timing gates, placed at the starting line, was used to determine players' ability to perform the COD test. The test was repeated twice, and the best performance of each test was recorded for analysis. A 6–8 min passive recovery was allowed between each trial.

*Agility T-Test.* The T-Test was performed according to *Raya et al. (2013)*. On the signal, the players ran as quickly as possible forward to the center cone (9.14 m), then turned and ran to the right cone (4.57 m), then ran to the far left cone (9.14 m), came back to the center cone (4.57 m), and finally ran or moved backward as quickly as possible to cross the start/finish line (9.14 m). The time to complete each trial was recorded in seconds by one set of the aforementioned dual-beam electronic timing gates, placed at the starting line. Players performed two attempts separated by at least 4–5 min of passive recovery, and the fastest one was recorded.

### Jump assessment

*Vertical jump test.* Vertical-jump height was measured using the VERTEC (Questtek Corp, Northridge, CA), following the protocol described by *Markovic et al. (2004)* Players began from a static standing position with their heels together and feet flat on the floor. While standing upright, they reached with their preferred arm to touch the zero vane (the non-preferred arm remained at their side). They then performed a countermovement jump by descending to a self-selected squat depth and jumping vertically as high as possible without a preparatory step or rebound, attempting to make contact with their fingers at the highest vane. The difference between the reach height and jump height was the performance score. Players performed three trials separated by at least 3 min of passive recovery, and the best result was recorded.

*The 5 jump test.* The five jump test is commonly used in field settings to assess players' horizontal jump performance, as an index of the lower limbs' explosive strength, by measuring the total distance covered in meters (*Bouhlel et al., 2007*). Players began the test with their feet together and chose which foot to lead with at the start. During the final stride, players were required to finish with their feet together. The total distance covered during the test was measured using a tape measure. Players performed three attempts, interspersed by 4–5 min of passive recovery, and the best attempt was recorded.

### Upper body strength and force assessment

*Seated medicine ball throw test.* The seated medicine-ball throw test is an easy, practical and valid measure of upper body strength (*Harris et al., 2011*). In a seated position on the gym mat with the knees bent at 45°, the players held a 1-kg medicine ball with both hands. At signal, the players lifted their torso and arms and threw the ball as far as possible. Three attempts were performed, interspersed by 4–5 min of passive recovery, and the best trial (in meters) was recorded for analysis.

*Hand grip test.* A handgrip dynamometer (Model 5030L1, Lafayette Instrument, USA) was used to measure hand grip force (*Roberts et al., 2011*). Players were asked to hold the dynamometer with the dominant hand, with the arm at a right angle and the elbow at the side of the body. Once ready, the players squeezed the dynamometer with maximum isometric effort, which was held for approximately 5 s. No other body movements were permitted. The result was taken from the digital display of the dynamometer to the nearest 0.1 kg. The value was reset to zero before each subsequent measurement. Two attempts were allowed, interspersed by 3 min of passive recovery, and the best result was recorded (*cf.* Supplementary File for more details).

*Trunk flexibility assessment.* The sit and reach test was used to assess trunk flexibility (*Castro-Piñero et al., 2009*). Sitting in front of a box (Baseline® Sit n' Reach Trunk Flexibility Box, Fabrication Enterprises Inc., USA), legs together and knees straight, the players flexed their trunk slowly with both arms outstretched and reached as far as possible with their fingers along the top of the box. Knee extension was monitored by the experimenter. Players performed two attempts, and the best result (in cm) was recorded for subsequent analysis.

*Balance assessment.* Balance was assessed using the Y-balance Test Kit™ (Move2Perform, Evansville, IN, USA; *Plisky et al., 2009*). The maximal reach distance (in cm) per leg and reach direction was used for further analysis. The composite score (%) was calculated and retained for analysis (*Plisky et al., 2009*). Players performed three data collection trials per leg and reach direction with 2-minute rest intervals.

### Serve accuracy and speed assessment

A serving precision test was used to measure serve accuracy (*Bartlett et al., 1991*) (Fig. 2). Players were asked to carry out five consecutive overhand serves (20 possible points) with a 15-second interval between each serve. Players were asked to perform their serves according to the position in the serving area they were more familiar with performing during their daily training. They were instructed to try to hit the ball at the highlighted areas (with values from two to four points) on the other side of the court in order to achieve the highest score. The ball impact represented the player's accuracy. Scoring zones are clearly labeled and easily interpretable, ensuring full reproducibility. If the ball hit outside the

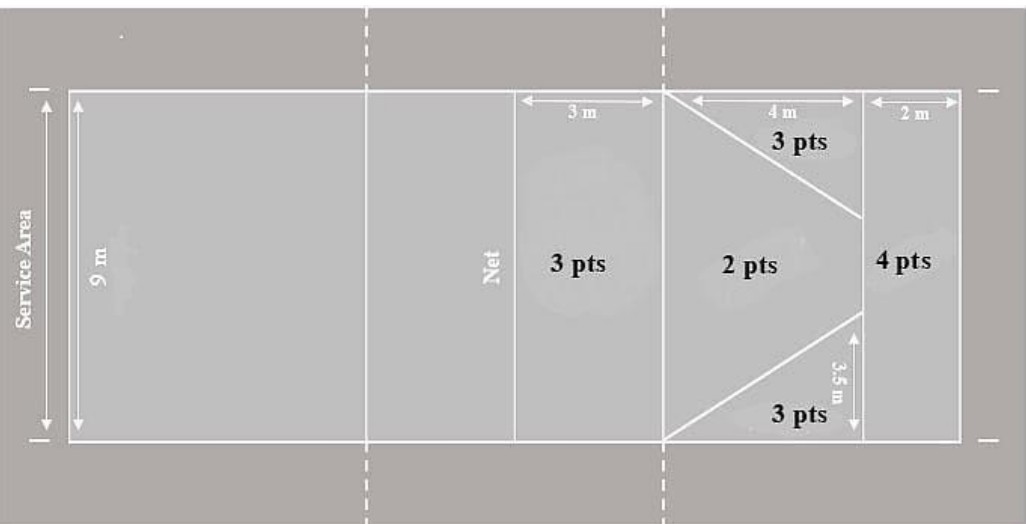

**Figure 2   Serve test diagram.**

designated zones or contacted the net, the players received no point. Two attempts were allowed, interspersed by 2 min of passive recovery, and the best score was recorded. *Serve speed* (km h$^{-1}$) was measured using a radar gun (model Globus Bushnell Radar, 101911, USA) with an accuracy of one km h$^{-1}$. Players stood behind the end line (service area), with the evaluator on the opposite side. Players served the ball to the other side twice, and the evaluator measured the speed of each serve. The fastest speed of the serve was retained for subsequent analyses.

## Statistical analyses

The statistical analyses were conducted using *IBM SPSS* software (*version 26.0; IBM Corp., Armonk, NY, USA*). After normality checks using the Shapiro–Wilk test, all data were expressed as mean ± standard deviation (SD), except for body fat, arm length, hand length, leg length, agility, COD, balance and serve speed which were expressed as median ± interquartile range (normality not assumed). We used Levene's test to check the homogeneity of variance, and scatter plots to test the linearity assumption. The reliability of the physical tests (*i.e.,* intraclass correlation coefficient (ICC)), standard error of measurement [SEM]) was calculated using the results from the familiarization sessions and those obtained during the second and third visits. Sex differences were analyzed by unpaired *t*-test or by Mann–Whitney test, accordingly. Pearson's r was used as effect size (ES) to estimate the magnitude of difference (0.1 to 0.29: small; 0.3 to 0.49: medium; 0.5 and higher: large) (*Cohen, 1988*). According to the normality of the distribution, the relationships between anthropometric/physical variables and serve accuracy/serve speed were established using Spearman's correlation coefficient or Pearson's correlation coefficient (r). Furthermore, Cook's distance (Di) was used to identify influential data points in the analyzed regression models (Di > 0.85) (*McDonald et al., 2002*). The stepwise multivariate linear regression method was used to identify the

**Table 2 Physical and technical performance of adolescent U14 boys ($n = 25$) and girls ($n = 25$).**

| Variables | Boys | Girls | P | Effect size | ICC | SEM (%) |
|---|---|---|---|---|---|---|
| 10 m Sprint (s)[*] | 3.0 ± 0.5 | 3.0 ± 0.4 | NS | – | 0.996 | 0.252 |
| COD (s)[**] | 12.2 ± 1.9 | 13.3 ± 2.2[‡] | 0.002 | 0.432 | 0.999 | 0.189 |
| Agility (s)[**] | 12.7 ± 1.5 | 12.6 ± 2.1 | NS | – | 0.985 | 2.939 |
| Vertical jump (cm)[*] | 37.3 ± 4.3[†] | 31.4 ± 5.5 | <0.0001 | 0.513 | 0.998 | 1.878 |
| Horizontal jump (m)[*] | 8.7 ± 1.3 | 7.9 ± 1.7 | NS | – | 0.999 | 0.221 |
| Upper limb strength (cm)[*] | 4.4 ± 0.8[†] | 3.7 ± 0.7 | 0.002 | 0.422 | 0.989 | 0.944 |
| Hand grip force (kg)[*] | 28.4 ± 4.5 | 28.4 ± 5.6 | NS | – | 0.987 | 6.499 |
| Trunk flexibility (cm)[*] | 6.1 ± 2.9 | 6.3 ± 2.7 | NS | – | 0.954 | 9.953 |
| Balance (%)[**] | 95.1 ± 4.6[†] | 83.7 ± 18.7 | <0.0001 | 0.73 | 0.997 | 5.915 |
| Serve accuracy[*] | 14.8 ± 2.7 | 13.3 ± 2.9 | NS | – | 0.983 | 5.476 |
| Serve speed (km h$^{-1}$)[**] | 42.0 ± 7.0[†] | 33.0 ± 4.5 | <0.0001 | 0.841 | 0.985 | 15.431 |

Notes.
[*]Values are expressed as mean ± SD.
[**]Values are expressed as median ± interquartile range.
ICC, intraclass correlation coefficient; SEM, standard error of measurement, COD: change of direction.
[†]Significantly different from girls.
[‡]Significantly different from boys.

predictors of serve accuracy and serve speed after checking the regression assumptions (homoscedasticity, multicollinearity and normal distribution residual). The standardized beta correlation coefficient and the coefficient of determination $R^2$ were used to assess the quality of fit of the model. Multicollinearity was assessed using the Variance Inflation Factor (VIF). A VIF > 10 and tolerance < 0.10 were considered indicative of multicollinearity. The significance level was established at $p \leq 0.05$.

# RESULTS

## Preliminary analysis of the data

All selected variables reached an excellent level of reliability (Tables 1 and 2; ICC > 0.90). Multicollinearity was tested, and variables with VIF > 10 or tolerance < 0.10 were excluded from the regression models (see footnotes in Tables 3, 4 and 5). Except for body fat and balance, Levene's test showed equal variance across samples, and the oval shape of scatter plots showed linearity of the data.

## Anthropometric attributes

Table 1 represents the anthropometric characteristics of the participants. In terms of PHV, boys showed faster upward growth in their stature, lower body fat, and higher leg length than girls (all $p < 0.001$) (Table 1).

## Physical and technical performance

Boys performed significantly better than girls for COD time ($p < 0.01$), vertical jump ($p < 0.0001$), upper body strength ($p < 0.005$), balance, and serve speed (both $p < 0.0001$) (Table 2). However, there were no differences between sexes in 10-m linear sprint, agility $t$-test, horizontal jump, hand grip force, trunk flexibility, and serve accuracy performance (Table 2).

**Table 3  Multiple linear regression analysis of the anthropometric variables on serve accuracy (23 boys and 21 girls) and speed (23 boys and 23 girls) for all participants.**

| Dependent variable | Independent variables | B | Beta | t | p |
|---|---|---|---|---|---|
| Serve accuracy | (Constant) | −49.049 | 0.510 | −7.087 | 0.0001 |
| | Lower limb length | 0.554 | 0.490 | 5.808 | 0.0001 |
| | Hand length | 0.930 | | 5.574 | 0.0001 |
| Serve speed | (Constant) | −123.900 | 1.209 | −9.670 | 0.0001 |
| | Height | 1.340 | −0.540 | 11.783 | 0.0001 |
| | Arm length | −0.745 | | −5.265 | 0.0001 |

Notes.

Serve accuracy: $R = 0.889$, $R^2 = 0.789$, standard error of estimation $= 1.335$, $F = 76.870$ ($p < 0.0001$). Excluded factor due to multicollinearity: Height (VIF $= 16.218$; Tolerance $= 0.062$).

Serve speed: $R = 0.890$, $R^2 = 0.792$, standard error of estimation $= 2.688$, $F = 82.076$ ($p < 0.0001$). Excluded factor due to multicollinearity: Lower limb length (VIF $= 16.404$; Tolerance $= 0.058$).

Significant variables in univariate analyses were included and stepwise multiple linear regression was used to predict the main factors influencing serve accuracy and speed. Assumed predictors: height, arm length, lower limb length and hand length.

**Table 4  Multiple linear regression analysis of the anthropometric variables on serve accuracy and speed in boys and girls.**

| Dependent variable | Independent variables | Boys | | | | Girls | | | |
|---|---|---|---|---|---|---|---|---|---|
| | | B | Beta | t | p | B | Beta | t | p |
| Serve accuracy | (Constant) | −111.387 | 0.949 | −12.226 | 0.0001 | −14.563 | 0.806 | −3.116 | 0.006 |
| | Height | 0.768 | | 13.827 | 0.0001 | 1.641 | | 5.937 | 0.0001 |
| | Hand length | | | | | | | | |
| Serve speed | (Constant) | 11.898 | 0.831 | 2.617 | 0.016 | −36.765 | 0.853 | −3.933 | 0.001 |
| | Hand length | 1.872 | | 6.834 | 0.0001 | 0.444 | | 7.500 | 0.0001 |
| | Height | | | | | | | | |

Notes.

Serve accuracy: Boys ($n = 23$): $R = 0.949$, $R^2 = 0.901$, standard error of estimate $= 0.878$, $F = 191.192$ ($p < 0.0001$); Excluded factor due to multicollinearity in boys: Lower limb length (VIF $= 10.736$; Tolerance $= 0.092$). Girls ($n = 21$): $R = 0.806$, $R^2 = 0.605$, standard error of estimate $= 1.716$, $F = 35.246$ ($p < 0.0001$).

Serve speed: Boys ($n = 23$): $R = 0.831$, $R^2 = 0.690$, standard error of estimate $= 2.081$, $F = 46.710$ ($p < 0.0001$). Girls ($n = 23$): $R = 0.853$, $R^2 = 0.728$, standard error of estimate $= 1.443$, $F = 56.246$ ($p < 0.0001$); Excluded factor due to multicollinearity in girls: Lower limb length (VIF $= 13.994$; Tolerance $= 0.071$).

All possible variables were included, and stepwise multiple linear regression was used to predict the influencing factors of serve accuracy and speed among players with different sexes. Assumed predictors: height, arm length, lower limb length and hand length. This approach was chosen for its ability to identify the most influential factors among a large set of potential predictors, providing a comprehensive understanding of the factors affecting serve accuracy and speed in adolescent volleyball players.

## Correlation analysis

There were positive correlations between height, arm length, lower limb length and hand length, and serve accuracy and serve speed in both boys and girls (range: $r = 0.497$ to $0.789$; all $p < 0.05$) (Figs. 3A, 3B, 4B, and 4C).

In addition, serve accuracy and serve speed were also significantly correlated with all physical outcomes (range: $r = -0.402$ to $0.908$; all $p < 0.05$), except the 10-m sprint time, which was only significantly correlated with serve speed in girls ($r = -0.403$; $p < 0.05$) (Figs. 3A, 3B, 4B, and 4D).

## Multiple linear regressions

For anthropometric attributes, the multiple linear regression analysis showed that lower limb length and hand length were the main factors influencing serve accuracy (78.9%), while height and arm length were the main factors influencing speed serve (79.2%) (Table 3). The equations for predicting serve performance for all participants were as follows:

Serve Accuracy $= -49.049 + 0.554 \times$ Lower Limb Length $+ 0.930 \times$ Hand Length

**Table 5** Multiple linear regression analysis of the selected physical performance on serve accuracy (25 boys and 24 girls) and speed (24 boys and 25 girls) for all participants.

| Dependent variable | Independent variables | B | Beta | t | p |
|---|---|---|---|---|---|
| Serve accuracy | (Constant) | 9.472 | −0.268 | 1.908 | 0.063 |
| | COD | −0.625 | 0.372 | −2.656 | 0.011 |
| | Trunk Flexibility | 0.394 | 0.262 | 4.239 | <0.0001 |
| | Balance | 0.072 | | 2.868 | 0.006 |
| | Hand grip force | 0.139 | 0.246 | 2.665 | 0.011 |
| Serve speed | (Constant) | 0.471 | 0.515 | 0.118 | 0.906 |
| | Balance | 0.285 | 0.415 | 5.023 | <0.0001 |
| | Upper limb strength | 2.896 | | 4.040 | <0.0001 |

**Notes.**
Serve accuracy: $R = 0.862$, $R^2 = 0.743$, standard error of estimation $= 1.519$, $F = 31.722$ ($p < 0.0001$). Excluded factor due to multicollinearity: Vertical jump (VIF $= 13.994$; Tolerance $= 0.071$).
Serve speed: $R = 0.839$, $R^2 = 0.705$, standard error of estimation $= 3.176$, $F = 54.880$ ($p < 0.0001$).
Significant variables in univariate analyses were included and stepwise multiple linear regression was used to predict the related influential factors of serve accuracy and speed. Assumed predictors: change of direction (COD), vertical jump, agility, horizontal jump, upper limb strength, hand grip force, trunk flexibility and balance.

Serve Speed $= −123.900 + 1.340 \times$ Height $−0.745 \times$ Arm Length.

When pooled by sex, the main factor influencing serve accuracy was height in boys (90.1%) and hand length in girls (65.0%), and the opposite was true for serve speed (Table 4). Thus, the equations for predicting serve performance were:

- For boys:

Serve Accuracy $= −111.387 + 0.768 \times$ Height
Serve Speed $= 11.898 + 1.872 \times$ Hand Length.

- For girls:

Serve Accuracy $= −14.563 + 1.641 \times$ Hand Length
Serve Speed $= −36.765 + 0.444 \times$ Height.

For physical performances, Table 5 shows that COD, trunk flexibility, balance and hand grip force were the main factors influencing serve accuracy in all players (74.3%), while balance and upper limb strength were the main factors influencing serve speed (70.5%) (Table 5). The equations for predicting serve performance for all participants were as follows:

Serve Accuracy $= 9.472–0.625 \times$ COD $+ 0.394 \times$ Trunk Flexibility $+ 0.072 \times$ Balance $+ 0.139 \times$ Hand Grip Force
Serve Speed $= 0.471 + 0.285 \times$ Balance $+ 2.896 \times$ Upper Limb Strength.

When analysed separately by sex, Table 6 shows that the main factors influencing serve accuracy were trunk flexibility, balance and COD among the boys (71.7%), while trunk flexibility and balance were the main factors among the girls (82.0%). Thus, the equations for predicting serve accuracy were:

For boys: Serve Accuracy $= −16.297–0.785 \times$ COD $+ 0.287 \times$ Trunk Flexibility $+ 0.408 \times$ Balance
For girls: Serve Accuracy $= 1.426 + 0.807 \times$ Trunk Flexibility $+ 0.086 \times$ Balance.

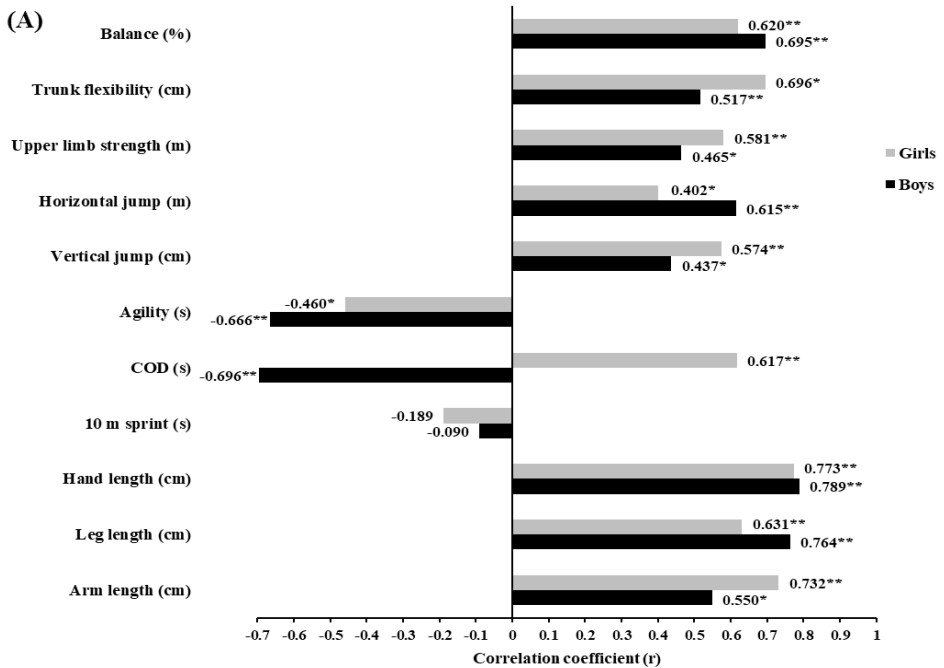

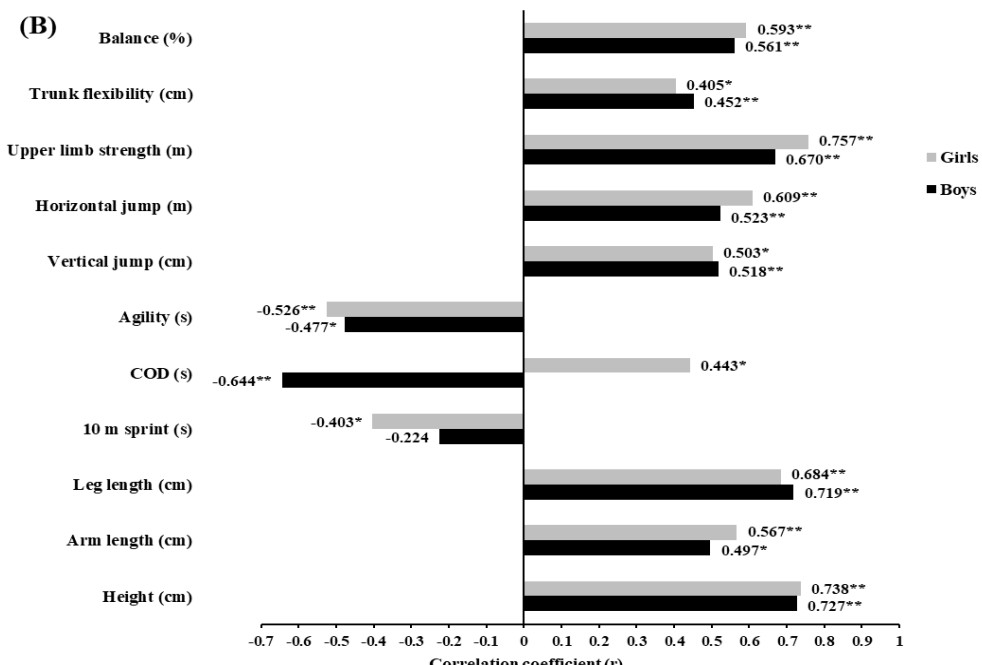

**Figure 3** (A) Relationship between serve accuracy and the independent variables in boys ($n = 25$) and girls ($n = 25$); (B) Relationship between serve speed and the independent variables in boys and girls; **COD: change of direction.** *significant at $p < 0.05$; **significant at $p < 0.001$.

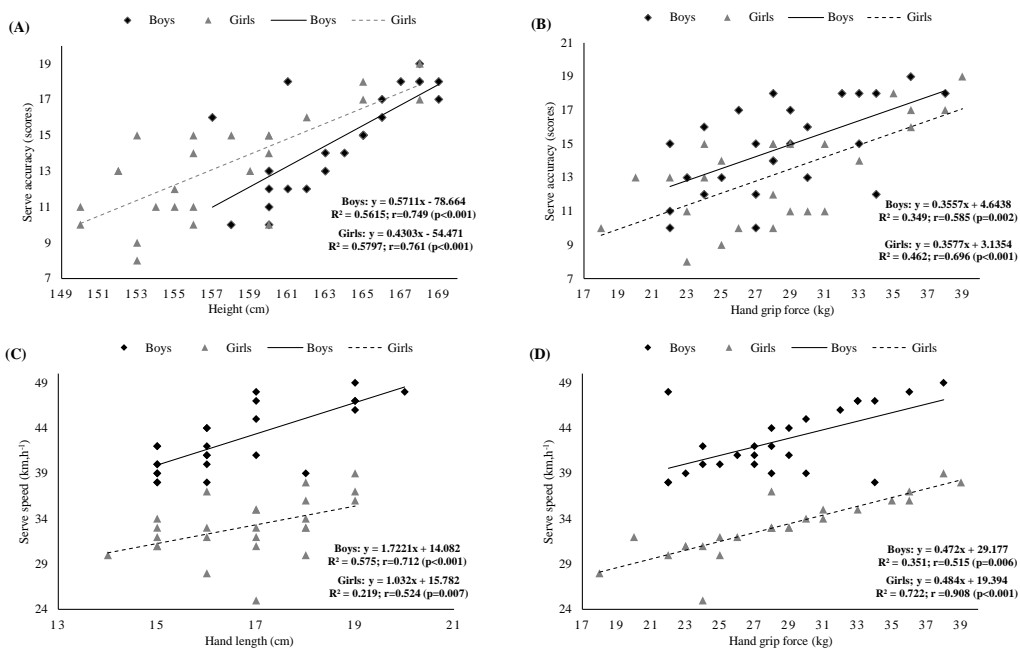

**Figure 4** (A) Relationship between serve accuracy and height in boys and girls; (B) Relationship between serve accuracy and hand grip force in boys and girls; (C) Relationship between serve speed and hand length in boys and girls; (D) Relationship between serve spe.

**Table 6** Multiple linear regression analysis of the selected physical performances on serve accuracy and serve speed in boys and girls.

| Dependent variable | Independent variables | Boys | | | | Girls | | | |
|---|---|---|---|---|---|---|---|---|---|
| | | B | Beta | t | p | B | Beta | t | p |
| Serve accuracy | (Constant) | −16.297 | −0.335 | −1.272 | 0.217 | 1.426 | 0.701 | 0.657 | 0.518 |
| | COD | −0.785 | 0.308 | −2.429 | 0.024 | 0.807 | 0.314 | 2.890 | 0.0001 |
| | Trunk Flexibility | 0.287 | 0.486 | 2.459 | 0.023 | 0.086 | | 6.450 | 0.009 |
| | Balance | 0.408 | | 3.744 | 0.001 | | | | |
| Serve speed | (Constant) | −12.897 | 0.288 | −1.140 | 0.268 | 19.394 | 0.850 | 10.721 | <0.0001 |
| | Hand grip force | 0.228 | 0.398 | 1.769 | 0.092 | 0.484 | | 7.728 | <0.0001 |
| | Balance | 0.422 | 0.422 | 3.336 | 0.003 | | | | |
| | Upper limb strength | 1.856 | | 2.786 | 0.011 | | | | |

**Notes.**

Serve accuracy: Boys ($n = 25$): $R = 0.847$, $R^2 = 0.717$, standard error of estimate = 1.541, $F = 17.770$ ($p < 0.0001$); Girls ($n = 24$): $R = 0.905$, $R^2 = 0.820$, standard error of estimate = 1.296, $F = 47.767$ ($p < 0.0001$).

Serve speed: Boys ($n = 24$): $R = 0.878$, $R^2 = 0.770$, standard error of estimate = 1.792, $F = 22.347$ ($p < 0.0001$); Girls ($n = 25$): $R = 0.850$, $R^2 = 0.722$, standard error of estimate = 1.713, $F = 59.716$ ($p = 0.008$).

All possible variables were included, and stepwise multiple linear regression was used to predict the related influential factors of serve accuracy and speed among players with different sexes. Assumed predictors: change of direction (COD), vertical jump, agility, horizontal jump, upper limb strength, hand grip force, trunk flexibility and balance.

Serve speed was mainly influenced by hand grip force, upper limb strength and balance among boys (77.0%), while hand grip force was the main factor affecting serve speed in girls (72.2%) (Table 6). The serve speed could thus be predicted using the following equations:

For boys: Serve Speed = −12.897 + 0.228 × Hand Grip Force + 0.422 × Balance + 1.856 × Upper Limb Strength

For girls: Serve Speed = 19.394 + 0.484 × Hand Grip Force.

## DISCUSSION

The present study showed sex-related differences in body fat and leg length between boys and girls who play volleyball. These differences were also highlighted in physical performances, with better performance in boys for COD, vertical jump, upper limb strength, balance and serve speed. These findings are significant in the field of sports science and volleyball as they provide insights into the factors influencing serve accuracy and speed in adolescent volleyball players. For anthropometric attributes, the regression analysis revealed that the main factor influencing serve accuracy was height for boys and hand length for girls, and the opposite was true for serve speed. This understanding can help coaches and trainers tailor their training programs to the specific needs of male and female players. For physical performances, the regression analysis showed the importance of COD, trunk flexibility, balance and hand grip force for serve accuracy and speed in boys and girls, further contributing to our understanding of the key performance factors in volleyball.

### Anthropometric attributes and correlation analysis

Boys showed faster upward growth in their stature. This was accompanied by higher height and longer leg length. In addition, boys have lower body fat than girls. These results align with those previously reported by other investigations (*Albaladejo-Saura et al., 2022*; *Kozlenia et al., 2024*; *Zwierko et al., 2022*). These morphological differences could be explained by faster pubertal growth, and associated hormonal changes. These changes are mainly related to higher testosterone secretion in boys, which increases bone and muscle mass (*Albaladejo-Saura et al., 2022*), while girls are characterized by larger estrogen and progesterone secretion, leading to increases in fat mass (*Zwierko et al., 2022*). These changes are likely accompanied by higher physical performance in boys, as found in the present study for COD time, vertical jump, upper limb strength, and serve speed. This agrees with previous findings (*Albaladejo-Saura et al., 2022*; *Kozlenia et al., 2024*; *Zwierko et al., 2022*). *Zwierko et al. (2022)* pointed out the significant central role of maturation for COD performance in elite adolescent volleyball girls. Sex differences are attributed to hormonal changes, implicating more considerable testosterone secretion in boys, and in parallel with greater estrogen and progesterone liberation in girls, leading to differences in body composition, with larger muscle mass in boys, and higher fat mass in girls.

Additionally, moderate to strong correlations were found in the present investigation between height, leg length, arm length, and hand length and serve accuracy and speed in both sexes. These correlations highlight the significant link between body span and serve efficiency. A large wingspan probably allows the player to better control the ball and the space in front, thereby improving serve success. Our study showed that maturation, body length and composition are important for serve efficiency. The regression analysis supported this observation. Indeed, taking all subjects together (boys and girls), we found that leg length and hand length were the main factors influencing serve accuracy (78%). Height and arm length were the main factors affecting serve speed (79%). Once again, body span plays a significant role in serve efficiency in our adolescent volleyball players. When considering sex, the main factor affecting serve accuracy and serve speed was height

for boys and hand length for girls, and the opposite was true for serve speed. Our data showcase the importance of hand length in the effectiveness of the serve in volleyball.

To the best of our knowledge, there are no studies that have examined the relationships between these aspects. In a recent study, *Kozlenia et al. (2024)* examined the relationship between lower leg length, fat mass index, and COD or reactive agility in male and female volleyball players. The study showed a negative correlation between relative leg length and COD time, positive correlations between BMI and COD and body fat index and agility in boys. The authors highlighted the importance of fat mass as a limiting factor for COD and agility performance in volleyball players (*Kozlenia et al., 2024*). *Tsoukos et al. (2019)* also reported that body height, BMI, and jump height are decisive factors for the selection of elite young female volleyball players.

## Physical performance and correlation analyses

The present study showed that boys performed better than girls in COD, vertical jump, upper limb strength, balance, and serve speed. Our results are in agreement with those previously reported in the literature (*Albaladejo-Saura et al., 2022*; *Kozlenia et al., 2024*; *Zwierko et al., 2022*). Specifically, these studies reported better physical performance in jump height, upper limb strength, agility, and reactive agility in boys compared to girls (*Albaladejo-Saura et al., 2022*; *Kozlenia et al., 2024*; *Zwierko et al., 2022*). The impact of sex on performance has been attributed not only to hormonal changes but also to neuromuscular characteristics (*Márquez et al., 2017*). *Márquez et al. (2017)* reported superior jump performance, with higher peak ground reaction forces, greater stiffness, and greater electromyography activity in the tibialis anterior and rectus femoris in males compared to females during the landing phase.

In our study, the correlation analyses demonstrated significant correlations between all physical performance variables and both serve accuracy and speed in boys and girls. While significant correlations (*e.g.*, $r \approx 0.40$) were observed between some performance metrics and serve outcomes, these associations should be interpreted with caution. Such values indicate a modest relationship and suggest that additional variables (*e.g.*, technical, cognitive, or tactical) may also influence serve performance. The multiple regression analysis showed that COD, trunk flexibility, and balance were the main factors affecting serve accuracy for all participants. Serve speed was mainly influenced by balance and medicine ball throw performance. Considering sex, COD, trunk flexibility, balance and hand grip force were the main factors influencing serve accuracy in boys (~74%), while only trunk flexibility and balance were the main factors affecting serve accuracy in girls (82%). Regarding serve speed, it was mainly influenced by hand grip force, medicine ball throw distance, and balance in boys (77%), and hand grip force for girls (72%). These results clearly emphasize the importance of trunk flexibility, balance, and upper limb force (*i.e.,* medicine ball throw and hand grip tests) for serve accuracy and speed. Thus, these physical aspects should be developed and incorporated into the training process to optimize serve performance. However, in adult female volleyball players competing in the Serbian Second League, height and body mass were positively related to explosive power (counter movement jump with arms), while muscle mass percentage was also positively

related to agility $T$-test time (*Ilić, Stojanović & Mijalković, 2023*), indicating that higher muscle mass proportion was associated with slower performance in the test. In contrast, *Pawlik et al. (2022)* assessed twelve 12–13-year-old youth female players from the Lower Silesian Regional Volleyball Team (one of the top three Polish U13 regional squads) and found that upper- and lower-limb strength measures (hand grip force, medicine ball throw, and shoulder internal-rotator peak torque) were the strongest positive correlates of serve reception efficiency. These findings suggest that different physical attributes may underpin performance depending on age and competitive level, with agility and flexibility playing a more prominent role in youth players, while strength and muscle mass may become more influential in adults.

While our study highlights the relevance of morphological and physical performance in serve efficiency in young volleyball players of both sexes, some limitations should be acknowledged. The study was conducted in homogeneous sample of Palestinian national youth players. This sample selection could limit the generalizability of our findings, despite the importance of our data in training and selection processes. In addition, the number of participants from each sex was relatively low, and the study was conducted on only one age group (under 14 years old). Thus, it would be beneficial to increase the number of participants and study other categories to better understand the impact of the development process on morphological, physical and technical variables. Other anthropometric attributes, such as muscle mass, could also be of interest when assessing the relationships between anthropometric, physical and technical aptitudes. Although the regression models were built using significant predictors and validated assumptions, the modest sample size, especially when stratified by sex, combined with the use of stepwise regression procedures may have introduced a risk of overfitting. This is particularly relevant given the high $R^2$ values observed in several models (*e.g.*, $R^2 > 0.80$), which, while indicating strong associations, may overestimate predictive accuracy in small samples. Additionally, the absence of internal validation methods (*e.g.*, cross-validation) limits the ability to assess model generalizability. Therefore, these findings should be considered preliminary and interpreted with caution. Future research using larger, independent cohorts and incorporating validation techniques is recommended to confirm the robustness and predictive value of these models.

## CONCLUSIONS

The present study showed the relevance of body span (height, leg, arm, and hand lengths) to serve accuracy. For all subjects, serve accuracy was influenced by COD, trunk flexibility, and balance. Conversely, serve speed was mainly affected by upper limb strength and balance. Considering sex, COD, trunk flexibility, balance, and hand grip force were the main factors affecting serve accuracy in boys, while trunk flexibility and balance were important for serve accuracy in girls. Serve speed was affected mainly by hand grip force, upper limb strength, and balance in boys, while hand grip force was the main factor for serve speed in girls. Overall, body span, force, flexibility, and balance were the main contributors to serve efficiency in adolescent players. When designing training programs

or selecting athletes, coaches and fitness trainers may benefit from considering a range of physical attributes, including coordination, muscle strength, flexibility, and balance. While we observed associations between variables and serve accuracy and speed, it is important to note that these relationships are correlational and do not imply causation. Nonetheless, coaches and trainers may consider body span (*e.g.*, height, arm, and hand length), upper limb strength, balance and trunk flexibility in different practical applications like player development and selection processes.

## ACKNOWLEDGEMENTS

The authors thank all the parents, coaches and players who participated in the study. The authors also thank Dr Zachary J. Crowley-McHattan for the final revision of the article.

### Funding

The authors received no funding for this work.

### Competing Interests

Yung-Sheng Chen is an Academic Editor for PeerJ.

### Author Contributions

- Rani Asfour conceived and designed the experiments, performed the experiments, analyzed the data, prepared figures and/or tables, and approved the final draft.
- Hamza Marzouki conceived and designed the experiments, analyzed the data, prepared figures and/or tables, authored or reviewed drafts of the article, and approved the final draft.
- Ibrahim Ouergui conceived and designed the experiments, authored or reviewed drafts of the article, and approved the final draft.
- Jabeur Methnani conceived and designed the experiments, authored or reviewed drafts of the article, and approved the final draft.
- Khalifa Al-Jadidi conceived and designed the experiments, authored or reviewed drafts of the article, and approved the final draft.
- Majid Al-Busafi conceived and designed the experiments, authored or reviewed drafts of the article, and approved the final draft.
- Yung-Sheng Chen conceived and designed the experiments, authored or reviewed drafts of the article, and approved the final draft.
- Nabil Gmada conceived and designed the experiments, authored or reviewed drafts of the article, and approved the final draft.
- Ezdine Bouhlel conceived and designed the experiments, analyzed the data, authored or reviewed drafts of the article, and approved the final draft.

### Human Ethics

The following information was supplied relating to ethical approvals (*i.e.*, approving body and any reference numbers):

The Faculty of Medicine of Sousse (University of Sousse) granted Ethical approval to carry out the study (Ref: CEFMS 402/2024).

## Data Availability

The raw data for all the dependent variables of the study for both girls and boys are available in the Supplemental File.

## Supplemental Information

Supplemental information for this article can be found online at http://dx.doi.org/10.7717/peerj.19992#supplemental-information.

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
