# Peer review of "Sex-based influence of morphological and physical attributes on serve performance in adolescent volleyball players"

_PeerJ, doi:10.7717/peerj.19992_

## Round 0.1 · original submission · Minor Revisions

Overall, your research presents an interesting and suitable topic, with a strong design adopted. The reviewers were largely positive regarding these strengths of the work, among others; however, there were some areas that require further attention. In particular, writing aspects and clarity, strengthening the rationale within the introduction, and methodological/statistical considerations were raised as key concerns by the reviewers. I agree with these suggestions, so encourage you to consider the suggested revisions closely and address them via suitable reasoning and changes to the manuscript where deemed appropriate. In addition, please consider improving Figure 2 by keeping the number of decimal places consistent for all values and ensuring they do not overlap with each other nor the bars/lines in the graph. In addition, try to include relevant data that supports the validity/adoption of each test in this population from the existing literature where available.

**Language Note:** The review process has identified that the English language must be improved. PeerJ can provide language editing services - please contact us at [email protected] for pricing (be sure to provide your manuscript number and title). Alternatively, you should make your own arrangements to improve the language quality and provide details in your response letter. – PeerJ Staff

Reviewer 1 ·

Basic reporting

The manuscript is generally well-structured and presents the research in a logical and coherent manner. The figures and tables included are appropriate, well-labeled, and support the findings effectively. The authors have clearly stated that the study received ethical approval, and the procedures regarding participant consent are adequately described.

However, there are several issues with the English language that need to be addressed to ensure clarity and readability for an international audience. In particular, there are grammatical and syntactic errors that occasionally hinder comprehension. For instance, the phrase "sexual differences" in the abstract should be corrected to "sex differences," which is the appropriate terminology in the context of biological and performance comparisons. Similarly, phrases such as “motor skills’ changes” (line 57) and “may lead to varied proficiency levels” (line 61) should be revised to “changes in motor skills” and “may lead to varied proficiency levels,” respectively.

To improve the manuscript’s clarity and overall quality, I strongly recommend that the authors have the text carefully reviewed by a native English speaker or a professional language editing service with experience in academic scientific writing. Addressing these issues will significantly enhance the manuscript's readability and ensure that the scientific content is communicated effectively.

Experimental design

Strengths:

The sample includes male and female youth volleyball players matched by biological age (Pre-PHV), which is appropriate and relevant.

The methodology is sound and well-explained, particularly the testing battery and the control of extraneous variables (ambient temperature, time of day, warm-up procedures).

Concerns:

The rationale for the selection of specific motor tests and their linkage to serve performance could be better justified in the Introduction.

The testing protocol is well-documented, but it would benefit from a schematic figure summarizing the testing flow.

Validity of the findings

Strengths:

Results are clearly presented and statistically robust.

Use of R² values to describe the explanatory power of predictors is appreciated.

The inclusion of multiple physical parameters provides a holistic view of server performance correlates.

Areas for caution:

Some of the associations reported, especially those with moderate r values (e.g., r = 0.41), should be interpreted with caution.

Given the relatively small sample size (n = 50 total), overfitting of regression models is a potential issue, especially when stratifying by sex.

Additional comments

This study contributes meaningful insights into the sex-specific determinants of serve effectiveness in adolescent volleyball players. However, several improvements would enhance the scientific and editorial quality of the work:

Clarify the novelty: It’s stated (lines 63–66) that no prior studies have explored these relationships, but references to related works (like Pawlik et al. 2022) suggest partial prior evidence. Rephrase more accurately.

Improve justification in the Introduction: While the aim is clear, the link between morphological attributes and volleyball technique could be strengthened with neurophysiological or biomechanical literature. This would help better explain why arm length, hand size, or grip strength might influence serve accuracy/speed.

Statistical interpretation: The authors could consider adding a multivariate regression or stepwise model to identify the most critical predictors overall, rather than separate univariate R² per variable.

Figures and tables: Consider adding visual plots to show the relationships (e.g., scatter plots with trend lines) between serve performance and the most important predictors (e.g., hand length, grip strength, balance score).

I strongly recommend the authors cite the following paper to enhance the scientific depth of the introduction, particularly regarding neuromuscular determinants of performance: Moscatelli et al., Transcranial magnetic stimulation as a tool to investigate motor cortex excitability in sport. Brain Sciences, 11(4), 432.
This study provides neurophysiological grounding for the influence of physical and morphological traits on performance, by highlighting the relationship between motor cortex excitability and athletic skills. This perspective could be useful when discussing how certain body dimensions and strengths may support better neuromuscular efficiency in skilled actions like serving.

Reviewer 2 ·

Basic reporting

-

Experimental design

-

Validity of the findings

-

Additional comments

In my opinion, everything is fine.

·

Basic reporting

The manuscript fulfills most PeerJ requirements under Basic Reporting but would greatly benefit from professional language editing, minor structural adjustments in the Methods and Results sections, expanded figure explanations, and improved raw data annotation. The suggested changes aim to enhance clarity, accessibility, and reproducibility.

1.1 English language
The manuscript is generally understandable and written in technical English appropriate for a scientific journal. However, several grammatical inaccuracies, stylistic inconsistencies, and improper terminological uses affect the fluency and precision of the text. A thorough professional language editing is recommended to ensure clarity and coherence.

Comment 1: In Line 58, the phrase "motor skills’ changes" is grammatically incorrect. Please revise to "changes in motor skills" to improve clarity.

Comment 2: In Line 65, "fitnesss" contains a typographical error. Please correct it to "fitness."

Comment 3: In Line 423, the simultaneous use of "serve accuracy and precision" is redundant. "Accuracy" alone suffices, or if both terms are intended, please clearly define the distinction between them.

Comment 4: Throughout the manuscript (e.g., Lines 25–90, 331–343), there is inconsistent use of "boys/girls" and "male/female athletes." Please adopt a consistent terminology across the manuscript.

1.2 Intro & background
The introduction provides a well-structured and contextually appropriate background, supported by recent and relevant literature. The knowledge gap is clearly identified, and the research question is logically derived from the existing evidence.

Comment 5: While the background is sufficiently comprehensive, the distinct contribution of your study compared to Albaladejo-Saura et al. (2022) and Kozlenia et al. (2024) should be further emphasized around Lines 63–67 to reinforce the novelty of your work.

Comment 6: Consider briefly discussing the broader implications of sex-specific differences in volleyball training in Lines 70–77, to frame the importance of your investigation more prominently.

1.3 Structure
The manuscript adheres to PeerJ structural guidelines (Abstract, Introduction, Materials and Methods, Results, Discussion, Conclusion). Nevertheless, the Materials and Methods section is overly detailed, which impacts the overall readability.

Comment 7: The Methods section from Lines 91–250 could be condensed by summarizing standard procedures and referring to detailed protocols (e.g., handgrip, sit-and-reach, vertical jump) in the Supplementary Materials. This would improve clarity without sacrificing replicability.

Comment 8: Consider restructuring the Results section (Lines 281–328) by presenting descriptive statistics separately from inferential findings to improve logical flow and facilitate reader comprehension.

1.4 Figures
Figures are generally relevant, clear, and properly labelled. They support the main results effectively. However, the accompanying descriptions in the text are sometimes overly brief, which may hinder the full appreciation of the figures’ meaning.

Comment 9: In Lines 304–308, please expand the description of Figures 2A and 2B to explicitly highlight key correlations and trends. This will guide the reader through the graphical information more effectively.

Comment 10: In Figure 1 (Serve Test Diagram), please confirm that all distances and scoring zones are clearly labelled and easily interpretable, ensuring full replicability.

1.5 Raw data
The raw data appears to be appropriately provided and is consistent with PeerJ’s policies for transparency and reproducibility.

Experimental design

The study is methodologically sound and adheres to high ethical standards. Minor revisions are needed to address sample generalizability, reduce redundancy in the Methods section, and improve transparency regarding evaluator blinding and measurement reliability.

2.1 Original primary research
The study presents original primary research that fits well within the aims and scope of PeerJ.

2.2 Research question
The research question is clearly formulated, meaningful, and relevant to the scientific community. The study builds upon prior literature while targeting a specific gap concerning sex-specific determinants of volleyball serve performance in pre-PHV athletes.

Comment 11: In Lines 78–90, the authors present their objectives well. However, consider strengthening the research question by explicitly linking it back to the literature gaps mentioned in Lines 63–77 to create an even clearer logical progression.

Comment 12: In Lines 84–90, the hypotheses are stated but could benefit from clearer separation into primary and secondary hypotheses to enhance focus.

2.3 Technical & ethical standard
The experimental procedures demonstrate a high technical and ethical standard. Participants' inclusion/exclusion criteria, familiarization with tests, and ethical approvals are properly addressed. Nonetheless, some methodological limitations require acknowledgment.

Comment 13: In Lines 98–111, the selection of a homogeneous sample (Palestinian national team players) limits the generalizability of findings. Please explicitly discuss this limitation in the Discussion section.

Comment 14: In Line 117, the ethics approval is clearly cited. It would be beneficial to state explicitly that the procedures adhered to the Declaration of Helsinki, as is customary in human research studies.

2.4 Methods
The description of methods is highly detailed, allowing for full replication. However, excessive procedural repetition reduces readability and efficiency, suggesting the need for summarization and better organization.

Comment 15: From Lines 91–250, the testing procedures are extremely detailed. Standard protocols for commonly used tests (e.g., handgrip strength, sit-and-reach, 10-m sprint) should be briefly summarized, and detailed operational steps could be moved to a Supplementary File.

Comment 16: In Line 129, when describing the environmental conditions ("temperature: 19–22 ºC; humidity: 67–76%"), it would enhance rigor to specify whether these conditions were measured and monitored using standardized instruments.

Comment 17: In lines 95–96, please clarify whether the physical and technical tests were administered by blinded evaluators to reduce potential bias.

Comment 18: In Line 274, it is indicated that Cook’s Distance was calculated to assess influential points. Please state the threshold used to define outliers (e.g., Di > 0.85) earlier, ideally in the Statistical Analysis subsection.

Validity of the findings

The findings are robust and well-analyzed, but concerns about overfitting must be openly discussed. Caution should be applied when drawing practical conclusions from correlational data. Minor revisions are needed to reinforce the methodological transparency and appropriate interpretation of the results.

3.1 Impact and novelty
The manuscript appropriately focuses on methodological robustness rather than overstating novelty, as per PeerJ policy. The rationale for replication and extension of previous findings is sufficiently outlined.

Comment 19: In Lines 63–77, the authors successfully identify a gap in the literature. However, it would be beneficial to explicitly emphasize the value of replication studies in adolescent sports science to reinforce the scientific contribution of the present research.

3.2 Data
All data are properly reported and statistically analyzed. However, concerns regarding model robustness and potential overfitting due to high R² values and a relatively small sample size must be addressed to enhance the credibility of the findings.

Comment 20: In Lines 283–286, the manuscript mentions that multicollinearity was checked using VIF, but the threshold values for VIF (>10) and Tolerance should be specified explicitly to increase transparency.

Comment 21: In Lines 315–317 (Table 4 results), serve accuracy in boys is predicted with R² = 0.901, which is unusually high given the small sample (n=23). Please discuss the potential for overfitting in the Discussion section and the need for model validation.

Comment 22: In Lines 329–331, regression models for physical performances show similarly high R² values (up to 0.820). Again, a comment about the risk of overfitting and the limitations of the stepwise regression method should be included in the Discussion.

Comment 23: In Line 274, Cook’s Distance is mentioned. Please confirm whether any data points exceeded the threshold and how these were handled (e.g., exclusion, sensitivity analysis).

3.3 Conclusions
The conclusions are generally well-aligned with the research question and the reported findings. However, some statements are overly prescriptive and should be reframed more cautiously to avoid implying causality based on correlational data.

Comment 24: In Lines 423–435 (Conclusions), statements such as "Trainers should use specific exercises..." imply prescriptive recommendations not directly tested within the study. Please rephrase these suggestions in a more tentative manner (e.g., "Trainers may consider...").

Comment 25: In Line 424, "serve accuracy and precision" are mentioned together. Please correct this redundancy, retaining only "accuracy" unless a distinct conceptual differentiation is intended and explained.

Comment 26: In Lines 413–420 (Limitations paragraph), the manuscript briefly mentions the small sample size. Please also discuss the homogeneity of the sample (Palestinian youth players) as a factor limiting generalizability.

Additional comments

Dear Authors,

Thank you for submitting your manuscript to PeerJ. Your study addresses an original and relevant research question concerning the influence of morphological and physical parameters on serve effectiveness in adolescent volleyball players, with a specific focus on sex-based differences. The rationale is clear, the experimental design is rigorous, and the statistical analyses are generally appropriate. Your work represents a valuable contribution to the field of youth sports science and talent development. After a thorough evaluation according to PeerJ's editorial standards, I believe that the manuscript is well-structured and largely meets the journal's requirements. Nevertheless, I suggest minor revisions to further enhance the clarity, precision, and impact of your work.
Specifically, I recommend the following points for improvement:

1. Language and Style:
Please revise the manuscript for grammatical errors, typographical mistakes, and consistency of terminology (e.g., consistent use of "boys/girls" or "male/female athletes"). Ensure clarity and conciseness, particularly in the Introduction and Discussion sections. Correct the misuse of "precision" where "accuracy" is meant (e.g., in the Conclusion).

2. Materials and Methods:
Although the methods are described in sufficient detail for replication, the section is overly lengthy and contains redundancies. I recommend summarizing standard test descriptions and moving extended procedural details (e.g., test repetitions, rest periods, familiarization procedures) into Supplementary Materials.

3. Statistical Modeling and Interpretation:
While the statistical analyses are comprehensive, some regression models present unusually high R² values (>0.90), suggesting a potential overfitting issue. Please discuss this limitation openly in the Discussion section, and if possible, consider supporting your findings with an internal validation method (e.g., cross-validation).

4. Discussion and Conclusion:
Soften prescriptive statements (e.g., recommendations for training interventions) unless they are directly supported by interventional study data. Frame conclusions in a way that acknowledges the observational nature of your findings.

5. Limitations:
Explicitly address the limited sample size, the specific population studied (Palestinian U14 athletes), and the potential implications for the generalizability of the findings.

Overall, your manuscript is a strong contribution, and I am confident that with these adjustments, it will meet the high standards of PeerJ.

---

## Round 0.2 · Minor Revisions

Thank you for thoroughly addressing all previous revisions and satisfying the requests from all reviewers. The manuscript has been greatly strengthened, but I just have some minor final considerations before I can recommend it being accepted. Please see these minor revisions listed below:

#1 Abstract - please ensure all r values and r-squared values are reported to the same number of decimal places for consistency (currently as two or three decimal places) and also indicate whether the correlations and regressions mentioned were statistically significant or not.

#2 Introduction, line 87: Perhaps change to "...is a critical period for talent identification and athletic development" here.

#3 Introduction, line 90: Please add "volleyball" before "study" on this line to clearly indicate you are referring to the volleyball literature.

#4 Introduction, line 96: Please indicate if these previous relationships reported were significant or not here (and perhaps the magnitudes if relevant).

#5 Introduction, lines 119-122: This sentence seems unnecessary given you are describing what you did before your aim. Perhaps consider removing this sentence and simply jumping straight into your aim statements, which are in the following sentences.

#6 Methods, line 141: Please change "weight" to "body mass" here and throughout all instances given weight is a force and you are referring to body mass in kg.

#7 Methods, lines 144-145: No need to capitalise "Total" or "See" here.

#8 Methods, line 221: It is unclear if a traditional "vertical jump" or a "drop jump" protocol was used here. Can you perhaps provide a little more detail briefly here within the main document to clarify this point.

#9 Methods, line 229: Add "each" before "participant" here.

#10 Methods, line 238: Add "a" before "1 kg". Also, change "his/her" to "their" for gender-neutral language.

#11 Methods, line 240: Change "used for the statistical analysis" to "recorded" given you provide this statement for other tests for consistency. Check this usage throughout the methods section please for consistency.

#12 Methods: Please provide the model details for all equipment consistently when describing the tests given this is inconsistently done in the main document.

#13 Methods, line 246: Change "are" to "were" and remove "recorded in kilograms" given you state the unit later in the sentence.

#14 Methods, line 253: Change " participants flexed the trunk" to "participant flexed their trunk" given you use singular form throughout this section elsewhere.

#15 Methods, line 262: Change "two" to "2" for consistency.

#16 Methods, line 266: Throughout the methods you refer to "The participant" when describing what was done, but here you refer to "Players" - in this regard, check throughout the entire manuscript and ensure consistency in how you describe the methods when referring to participants. I would probably prefer use of "players" in the plural form each time, but I will leave this decision to you.

#17 Methods, line 271: You use the term "subject" on this line (and throughout the discussion).

#18 Methods, line 277: Where you refer to "km" per "h" you separate these units with a full-stop rather than a centre-aligned dot symbol, so please adjust this throughout the manuscript where mentioned.

#19 Methods, line 286: Add "range" after "interquartile" and also "mormality" is not spelt correctly.

#20 Results, line 320: Please be specific here and say "agility t-test" instead of "agility" given this is a different measure and add "performance" after "accuracy" at the end of this sentence.

#21 Results, line 326: You state "moderate to strong positive correlations" but what constitutes "moderate" and what constitutes" strong"? Please specify these in the statistical analyses section if you have specific magnitudes for different descriptors - otherwise, please refrain from using "descriptors" and simply present the r value ranges.

#22 Results, lines 331-333: Perhaps for clarity you could state that "all physical outcomes" were significantly correlated rather than "most", but where you indicate the range of r values in brackets, this is not accurate given some were negative, so please adjust this accordingly. Then, where you mention "negatively" in the next sentence, change this to "significantly" given all other correlations for 10-m sprint time were not significant and you show the direction in brackets within this sentence.

#23 Figure 3: Consider splitting this figure into two separate figures where one contains the bars showing correlation magnitudes and the other shows the scatterplots. At present, it is quite extensive and difficult to interpret given so much data are presented together.

#24 Discussion, line 385: "olleyball" is not spelt correctly.

#25 Discussion, line 459: Perhaps remove "some authors have shown that" given you only cite one study for this statement.

#26 Discussion, lines 462-465: This is not a complete paragraph given it is a single sentence. Was it meant to be include in the previous paragraph? Please adjust so it does not sit on its own.

#27 Conclusion, lines 501-503: Consider removing "incorporating exercises that develop coordination, strength, 502 flexibility, and balance, can enhance overall physical fitness and potentially improve serve performance in volleyball." given you already highlight the importance of these attributes. Instead, you could just state "Nonetheless, coaches and trainers might consider these attributes during athlete development and selection processes."

Reviewer 1 ·

Basic reporting

Ok

Experimental design

Ok

Validity of the findings

Ok

Additional comments

Ok

·

Basic reporting

The authors have adequately addressed all issues related to language clarity, structural organization, and figure/data presentation. The manuscript was revised by a native English speaker, and all key terminology inconsistencies have been corrected. Redundancies in the Methods section were removed, and standard procedures were moved to Supplementary Materials. Figure legends and references have been expanded for clarity, and raw data annotations were improved. Overall, the manuscript now meets PeerJ standards for transparency and reporting quality.

Experimental design

The revised manuscript presents a methodologically sound design, with improved justification for test selection and clearer articulation of the research hypothesis. A schematic representation of the protocol was added (Figure 1), evaluator blinding status was clarified, and reliability assessments were expanded. The addition of biomechanical and neuromuscular literature in the Introduction strengthens the theoretical rationale. The description of testing procedures is now streamlined yet fully replicable.

Validity of the findings

All major concerns regarding statistical modeling, overfitting, and interpretation of moderate correlations have been carefully addressed. The authors now openly acknowledge the sample size limitation, the absence of internal validation (e.g., cross-validation), and the use of stepwise regression. The conclusions have been reformulated to avoid prescriptive or causal claims, in line with the correlational nature of the study. The statistical assumptions (e.g., VIF thresholds, Cook’s Distance) are clearly described, and outlier management is transparently reported.

Additional comments

This revised version represents a substantial improvement over the initial submission. The authors have constructively engaged with all feedback, enhancing both the methodological transparency and scientific contextualization of their work. The study offers meaningful insights into youth volleyball performance and is now ready for publication.

---

## Round 0.3 · Minor Revisions

Thank you for making the requested changes to the manuscript, it has been greatly strengthened and is almost to a level acceptable for publication. I only have some very minor final points to make before recommending acceptance, which are presented below.

#1 Figure 3 - Please adjust the comma after the leading zero for correlation values next to each bar and on the axes to a full-stop ("." instead of "," each time). Please only capitalise the first word for each variable down the y-axis for consistency (unless it is a specific name for the test that is capitalised in the main document). Also, in Figure 4A, is the p value meant to be <0.001 for both equations rather than <0.0001 by convention?
#2 Line 470 - "significant" is not spelt correctly.
#3 Line 486 - please add "performance" or "time" after "agility T-test" here. Also, if it is a positive relationship and Agility T-test time is the measure, then this means higher muscle mass proportion was associated with worse performance times in the test? Can you please be more specific here to the performance measure and direction of the relationship.
#4 Lines 487-490 - It is not clear what population this study recruited (e.g., adult female volleyball players)? Also, please indicate the level of players in this sentence and the one prior if possible.
#5 Line 490 - Thank you for merging this paragraph, but it is unclear what these added studies mean in relation to your findings. So, please add a conclusive sentence highlighting what these additional findings mean - for instance, do they indicate other physical attributes/parameters might underpin performances in adult players compared to what you observed for youth players?
#6 Throughout - you seem to use different terms for the same thing - such as "attributes" and "parameters". Please try to be consistent so the reader can be aware you are referring to the same thing each time - probably use "attributes" throughout where applicable.
#7 Line 530 - Perhaps change the wording to "Nonetheless, coaches and trainers may consider these attributes in different practical applications like player development and selection processes". You could also be specific and indicate which attributes instead of "these attributes" in this sentence.
#8 The reference list is not formatted correctly, so needs finalising.

---

## Round 0.4 · accepted · Accept

Thank you kindly for addressing these revisions, the presented work is at a high standard and should make a meaningful contribution to this area.